# PN Administration in Critically Ill Children in Different Phases of the Stress Response

**DOI:** 10.3390/nu14091819

**Published:** 2022-04-27

**Authors:** Koen Joosten, Sascha Verbruggen

**Affiliations:** Intensive Care Unit, Department of Pediatrics and Pediatric Surgery, Erasmus MC Sophia Children’s Hospital, 3060 Rotterdam, The Netherlands; s.verbruggen@erasmusmc.nl

**Keywords:** parenteral nutrition, critical illness, infants, children, guidelines, amino-acids, lipids, micronutrients, pediatric intensive care, acute stress response

## Abstract

Nutritional support is an important part of the treatment of critical ill children and the phase of disease has to be taken into account. The metabolic stress response during acute critical illness is characterized by severe catabolism. So far, there is no evidence that the acute catabolic state can be prevented with nutritional support. The Pediatric ’Early versus Late Parenteral Nutrition’ (PEPaNIC) trial showed that withholding supplemental parenteral nutrition (PN) during the first week in critically ill children, when enteral nutrition was not sufficient, prevented infections and shortened the stay in the pediatric intensive care unit (PICU) and the hospital. A follow-up performed 2 and 4 years later showed that withholding parenteral nutrition (PN) also improved several domains of the neurocognitive outcome of the children. Current international guidelines recommend considering withholding parenteral macronutrients during the first week of pediatric critical illness, while providing micronutrients. These guidelines also recommend upper and lower levels of intake of macronutrients and micronutrients if PN is administered.

## 1. Introduction

Pediatric critical illness is characterized by neuro-endocrine, immunologic and me-tabolic alterations. These acute stress response alterations inhibit the normal developmental process to ensure survival [1]. The ultimate goal of nutritional support is to provide a sufficient amount of feeding in order to improve the short-term and long-term outcome. Both undernutrition and overfeeding have been associated with worse outcomes [2,3,4,5,6]. Critically ill infants and children are at risk for the development of nutritional macronutrient and micronutrient deficiencies, and therefore nutritional support is of great importance.

## 2. Acute Stress Response

The acute stress response to critical illness can be divided into three different phases: the acute, stable and recovery phase. Nutritional goals differ throughout the different phases of the disease [7]. The acute phase (Table 1) is characterized by (escalating) requirement of vital organ support and may last up to several days. In the stable phase, there is stabilization or weaning of vital organ support. Finally, the normalization of the stress response and clinical mobilization characterizes the recovery phase. Being aware of the metabolic changes during the different phases is essential to tailor metabolic and nutritional support with enteral nutrition (EN) and/or parenteral nutrition (PN).

## 3. Parental Nutrition in Critically Ill Children

PN is often started when enteral intake is not sufficient to reach recommended target nutritional intake goals. PN contains macronutrients (carbohydrates, amino acids, and lipids) as well as micronutrients (electrolytes, trace elements and vitamins). Previous recommendations for optimal timing, amount and composition of PN were based on very few studies in critical ill children, using intermediate or surrogate endpoints, such as inflammation markers and nitrogen balances [8]. Based on an expert consensus and these observational studies, PN was advised during all phases of critical illness to achieve nutritional goals [9].

These recommendations were reassessed after the findings of the Pediatric ’Early versus Late Parenteral Nutrition’ (PEPaNIC) randomized controlled trial (RCT) [10]. This large (*n* = 1440) multicenter RCT showed that withholding supplemental macronutrients via PN for seven days (late PN), as compared with initiating PN within 24 h after admission (early PN), improved the short-term outcome. Children within the late PN group had a lower incidence of newly acquired infections and a shorter length of pediatric intensive care unit (PICU) and hospital stay. The results were independent of confounders such as severity of illness, age and being undernourished upon admission. Secondary analyses of this study showed that term neonates and undernourished children benefited as well from withholding PN [11,12].

Importantly, a pre-planned follow-up performed 2 and 4 years later showed that withholding PN also improved several domains of the neurocognitive outcome of the children [13,14]. In parallel, a large group of matched healthy children was assessed during these follow-up moments. Overall, critically ill children were at risk of worse developmental outcomes as compared with matched healthy children [13,14,15]. At 2- and 4-years follow-up, former critically ill children had experienced more health problems and scored worse for growth, general intelligence, visuo-motor integration, alertness, verbal and nonverbal memory, working memory, parent-reported executive functioning, emotional and behavioral problems, and quality-of-life.

Recent studies have shown the amelioration of neuro-endocrine responses by withholding PN during the acute phase. A further reduction in plasma concentrations of the hormones of the thyroid axes was noticed for thyroid-stimulating hormone (TSH), total thyroxine (T4), triiodothyronine (T3), and the ratio of T3 to reverse T3. This was not seen in patients receiving early PN [16]. It is hypothesized that the inactivation of T4 to reverse T3 and T3 to T2, altering the T3/reverse T3 ratio, might be a beneficial adaptation during acute critical illness as a result of caloric restriction, which might be associated with improved outcome in critically ill children [16,17,18].

So far, no other RCTs have focused on optimal timing or the amount of PN in critically ill children. Only in two large observational studies were nutritional practices in relation to clinical outcomes studied [3,19]. In an observational study of 500 patients, it was found that the intake of a higher percentage of the prescribed dietary energy goal via EN was associated with improved 60-day survival. In this study, it was also remarkable that PN was associated with higher mortality [3]. In another observational study of 1844 patients, the number of patients who received treatment on 1 or more PICU days PN was provided in 378 (20%) patients. In 275 of these patients, PN was used as a supplement to insufficient EN. In this study, the association between lower mortality and a nutrient delivery strategy, achieving 60% of the prescribed goal by day 7, was seen with both with delivery of EN and EN + PN. Based on their results, the authors advocate the selective use of PN to supplement EN, if necessary, before day 7 after admission [19].

Based on recent literature, the Society of Critical Care Medicine/European Society of Intensive Care Medicine (SCCM/ESICM), European Society of Paediatric and Neonatal Intensive Care (ESPNIC) and European Society for Paediatric Gastroenterology Hepatology and Nutrition/European Society for Clinical Nutrition and Metabolism/European Society for Paediatric Research/Colorado Society for Parenteral and Enteral Nutrition (ESPGHAN/ESPEN/ESPR/CSPEN) have updated their guidelines and recommend considering withholding parenteral macronutrients during the first week of pediatric critical illness, while continuing to provide micronutrients [20,21,22].

## 4. Practical Approach to Nutritional Support with PN during the Course of Critical Illness

### 4.1. First Week of Admission: Parenteral Glucose and Micronutrient Administration

In the acute phase of illness, the energy requirements of the human body, and especially the brain, depend on glucose and ketones as major fuels. Plasma glucose levels are the net resultant of the exogenous glucose intake and endogenous glucose production (glycogenolysis and gluconeogenesis) minus the glucose utilization (oxidation or storage as glycogen and triglycerides). Glucose metabolism in neonates and children is highly modified during the different phases of critical illness. During the acute phase, protein catabolism cannot be modified with increasing glucose intake. Both hyperglycemia, which occurs frequently during the acute phase, and hypoglycemia are undesirable [23,24,25,26]. Therefore, the basis for glucose intake recommendation in the different phases of the critically ill neonate and child deserves a separate approach. Studies have shown that reduced glucose intake in critically ill infants safely lowered high blood glucose levels when increased endogenous glucose production was present [27,28].

In term newborn neonates, the amount of iv glucose administration should usually be gradually increased depending on the day of birth, whereas in infants and children iv glucose infusion can be increased depending on the phase of disease (Table 2). In clinical practice, this means that critically ill neonates and young children on admission will receive a solution with glucose. On the second day after admission, if no EN or an inadequate amount of EN can be supplied, it is recommended to supply a glucose infusion with micronutrients (electrolytes, vitamins and trace elements) (Table 3) [29].

Initial screening for hypo- and hyperglycemia is recommended to perform in all cri-tically ill children. Hyperglycemia with high plasma insulin concentrations is the result of insulin insensitivity that occurs during the acute phase of stress. Both insulin resistance and (relative) β-cell dysfunction play a role in the occurrence of hyperglycemia. Additionally, excessive glucose intake should be avoided because it may not only be responsible for hyperglycemia but also causes increased lipogenesis and fat tissue deposition, together with subsequent liver steatosis and the enhanced production of very-low-density lipoprotein (VLDL) triglycerides by the liver. Hyperglycemia > 8 mmol/L (>145 mg/dL) should be avoided, and it is recommended to treat repetitive blood glucose levels > 10 mmol/L (>180 mg/dL) with continuous insulin infusion [23]. Repetitive and/or prolonged hypoglycemia < 2.6 mmol/L (45 mg/dL) has to be avoided in all critically ill patients. The use of a stepwise algorithm to treat hyperglycemia with lowering glucose intake or treatment with insulin is recommended (Figure 1).

### 4.2. Micronutrients

The start of both EN and PN can influence micronutrient levels and shifts. Micronutrient depletions are not only frequently observed in critically ill children but also associated with impaired morbidity and mortality [30]. However, numerous knowledge gaps persist regarding the interpretation of serum levels of individual micronutrients during critical illness. Nor is it clear whether these low levels are actually caused by micronutrient depletion on cellular or tissue level or whether they have shifted between compartments. Moreover, the causality of associations between micronutrient levels and clinical outcomes is uncertain and thus no recommendations for (routine) supplementations can be made. Currently, no trials exist on the effect of micronutrient supplementation, but from a physiological perspective continuing to provide micronutrients during all phases of critical illness make is recommended. Due to a lack of evidence, the current recommendations are based upon expert opinion and dietary reference nutrient intake for healthy children without accounting for the phase of critical illness. Although the ESPGHAN/ESPEN/ESPR/CSPEN and ASPEN guidelines both acknowledge that recommended intakes can only be accomplished using individualized micronutrient products in critically ill children [31,32], the ESPGHAN/ESPEN/ESPR/CSPEN guidelines recommend also to use commercial products due to the lower risk for microbial contamination and compounding errors, which makes it difficult to tailor treatment to individual needs [33].

Serum levels are used most often to determine micronutrient status. Other methods used are intracellular depletion measurements, clinical signs of deficiencies, and dietary assessments. However, all methods have their pitfalls. Clinical signs are often not differentiated from critical illness itself, and for serum and intracellular measurements pediatric reference standards have been published recently [34]. Furthermore, as micronutrient levels can be significantly lowered during inflammation, such circumstances require careful interpretation of these measurements [35]. Table 3 shows the micronutrient provision protocol in Rotterdam, The Netherlands. 

## 5. Parenteral Glucose and Micronutrient Administration after the First Week

### Energy

Endogenous energy production can cover half to two-thirds of energy requirements during acute critical illness, irrespective of the exogenous energy provision. This means that in the acute phase, energy intake provided to critically ill children should not exceed resting energy expenditure.

Figure 2 depicts a conceptual cartoon of the energy requirements during the phases of critical illness. In addition, Figure 3 shows a practical protocol for initiating EN and PN in the first 2 weeks after admission.

No interventional studies so far have determined the precise energy intake related to improved outcomes in critically ill children. In a systematic review [38], followed by an observational study [39], it was observed that a minimum intake of 57 kcal/kg per day (and 1.5 g/kg per day of protein) was associated with positive nitrogen balance. No difference was made in these studies between children receiving EN and/or PN. However, a nitrogen or whole-body protein balance cannot be conceived of as a clinically relevant outcome measure because it has no predictive outcome concerning the preservation of muscle mass.

Energy requirements after the first week of admission to guide PN should be ideally measured by indirect calorimetry. There are a few studies in adults and one in critically ill children who have used indirect calorimetry to target nutritional therapy. A meta-ana-lysis of four prospective randomized trials in adults evaluating the benefit of isocaloric nutrition guided by indirect calorimetry showed a reduced 28-day mortality in critically ill adult patients in the intensive care unit (ICU). However, no difference in 90-day mortality and nosocomial infection rate could be shown [40]. In a study of 139 critically ill children, energy intake was compared with measured energy requirements. If the delivered energy was less than 90%, this was classified as underfeeding, and if it was above 110%, it was classified as overfeeding. In this study, patients who were overfed showed a significantly longer PICU and hospital stay compared to those who were adequately fed [41].

However, in most clinical settings there is a lack of availability of indirect calorimetry devices, which means that prediction equations to calculate resting energy expenditure (REE) have to be used. Reasonable values for REE in critically ill children can be derived from Schofield’s prediction equation [42] for REE using the actual weight (and, if possible, measured height) of the patient (Table 4).

In the recovery phase, REE values can further be used as a guide for determining energy requirements; the body experiences a significant increase in metabolic needs, with energy requirements increasing up to as much as 1.5–2 × REE [36]. If the critically ill child is fully dependent on PN, there is a maximum amount of PN that can be administered (Figure 2). This is much lower compared to the amount of energy that can be administered with EN.

## 6. Amino Acids

Critically ill children will experience a net negative protein balance, which may be expressed clinically by weight loss, negative nitrogen balance and skeletal muscle wasting. This may have deleterious effects on outcome. During the acute phase of critical illness, increased protein intake by EN or PN cannot reverse or prevent muscle protein breakdown, and there are clear indications that increasing protein (amino acid) intakes during acute critical illness is detrimental for muscle architecture and function [43].

Amino acid requirements depend on age, the phase of illness, and the route of administration. Amino acids requirements in PN are lower as compared with EN as they bypass the utilization by the gastro-intestinal tract [44]. Parenterally administered amino-acids in the first week after admission have been shown to be harmful in already low doses [45]. A secondary analysis from the PEPaNIC RCT showed that during the acute phase of critical illness, higher doses of parenterally administered amino acids were negatively associated with length of stay in the PICU, newly acquired infections and the duration of mechanical ventilation [45]. Already low doses of parenteral amino acids were found to be harmful, whereby a maximal risk of harm was reached with a median daily dose of 1.15 g/kg for children <10 kg, 0.83 g/kg for children 10–20 kg and 0.75 g/kg for children >20 kg. After the acute phase, muscle wasting often continues due to immobilization and undernourishment.

The ESPGHAN/ESPEN/ESPR/CPNN guidelines recommend that a minimum amino acid intake of 1.5 g/kg/d should be administered to stable-term infants to avoid a negative nitrogen balance, while the maximum amino acid intake should not exceed 3.0 g/kg/d. For infants from the 2nd month up to 3 years, a minimum amino acid intake of 1.0 g/kg/d should be administered, and the maximum should not exceed 2.5 g/kg/day, and in children older than 3 years, the maximum intake should not exceed 2.0 g/kg/day [46]. Whether early mobilization of a critically ill child would allow for the better incorporation of amino acids into muscle protein is unknown, and thus the impact of early mobilization on the recommended protein intake requires further investigation.

## 7. Lipids

Lipid metabolism is generally accelerated by illness and physiologic stress, and lipids are a prime source of energy. Secondary analyses of the PEPaNIC RCT showed that the effects of the administration of carbohydrates and lipids were neutral or even beneficial [9]. The PEPaNIC RCT has raised an important issue on the best time to provide PN support in critically ill children. This trial also does not allow one to differentiate between potential effects of different PN components [10]. So far, the updated guidelines have recommended considering withholding parenteral macronutrients, and thus lipids, during the first week of critical illness. Parenteral lipid intake is fundamental when PN is initiated after the first week of admission and accounts for 25–50% of the non-protein caloric intake. Providing lipids allows for a high energy supply without administering high doses of carbohydrates. The administration of fatty acids, with a minimum of linoleic acid intake of 0.1 g/kg/day, is essential to prevent essential fatty acid deficiencies [47]. It is currently recommended not to exceed a lipid intake of 4 g/kg/day and 3 g/kg/day via PN in infants and children, respectively. The provided dosage of lipids should not exceed the capacity for lipid clearance and should be lowered in the case of hyperlipidemia (serum triglyceride level is >265 mg/dL (>3.0 mmol/L) in infants > 400 mg/dL (>4.5 mmol/L) in children. Critical illness can result in the acceleration of the lipid metabolism, and higher levels of serum triglycerides.

The choice of intravenous lipid emulsions (ILEs) is influenced by several considerations, which include the composition of the ILE, the duration of PN, the setting (home vs hospital PN), age, disease conditions, and other factors. When prescribing ILEs, an understanding of the biological properties and of the fatty acid components is mandatory. As the fatty acid compositions of current ILEs cannot tailor specific individual clinical needs, the metabolic profiles and the specific requirements of the critically ill child should guide the prescription of the best-available ILEs. This not only to improve short-term outcomes such as healing and recovery but also to improve long-term outcomes such as growth and cognitive development.

In newborns and older children who are on short-term PN, pure soybean oil (SO) ILEs may provide a less balanced nutritional intake than composite ILEs. It is recommended that if PN is required longer than a few days, pure SO ILEs should no longer be used. Composite ILEs with or without fish oil (FO) should be the first choice of treatment [21,48]. In critically ill children with septic shock or other sepsis-associated organ dysfunction, supplementation with specialized lipid emulsions is not recommended [20].

Children with severe, unexplained thrombocytopenia serum TG concentrations have to be monitored, and a reduction in parenteral lipid dosage may be considered. Intestinal-failure-associated liver disease (IFALD), historically called PN-associated liver disease (PNALD) or PN-related cholestasis, reflects a heterogeneous liver injury consisting of cholestasis, steatosis, fibrosis and even cirrhosis. Prematurity and catheter-related bloodstream infections are known to be the most important factors involved in the development of IFALD. As part of measures to reverse IFALD in children, a discontinuation of SO ILE, a reduction of other ILE dosage and/or the use of composite ILE with fish-oil should be considered, together with the treatment and management of other risk factors. The use of pure fish-oil ILE is not recommended for general use in children but may be used for short-term rescue treatment in children with progression to severe IFALD.

## 8. Discussion

There is a scarcity of data concerning well conducted RCTs on PN in critically ill children. In a systematic review published in 2016, six small RCTs were identified that investigated the impact of a different dose or composition of PN in critically ill infants or children [48]. Four of these 6 studies investigated infants after cardiac surgery, and two investigated children with sepsis or after other major surgery, or burns, respectively. The focus of these few studies was on intermediate or surrogate endpoints that appeared to be beneficially affected by providing more or altered parenteral nutrition early during critical illness. As the studies included a limited number of children, all were statistically underpowered to detect a relevant effect on clinical endpoints. In 2016, the landmark PEPaNIC RCT study was published, which investigated if withholding PN during the first week in PICU improved outcome, as compared with early supplementation of insufficient enteral nutrition (EN) with PN. The PEPaNIC RCT showed not only significant differences in short-term outcomes at the PICU (less infections and accelerating recovery) but also in long-term outcomes for those not using early-PN. The two pre-planned follow-up studies of the PEPaNIC RCT, performed 2 and 4 years after randomization, showed that not using early PN improved clinically tested visuo-motor integration and parent-reported executive functioning and emotional/behavioral problems. Children who were between 29 days and 11 months old at time of PICU admission appeared to be most vulnerable to the harmful effects of early-PN on neurocognitive function and behavior [49]. The development of aberrant epigenetic changes has been identified as a plausible mechanism underlying the developmental impairments observed 2 years and 4 years after critical illness [50].

Based on the results of the PEPaNIC, the RCT international guidelines have been updated, and all guidelines recommend considering withholding parenteral macronutrients during the first week of pediatric critical illness, while continuing to provide micronutrients in children.

The secondary analysis of the PEPaNIC RCT has also some limitations such as insufficient power in some of the subanalyses, and the observational design of these analysis. Furthermore, because the PEPaNIC RCT was not designed to study the optimal start of starting PN on a specific day, the timing to start PN is still under debate. In line with this, the writers of the ESPGHAN/ESPEN/ESPR/CSPEN guidelines for PN have come up with a consensus about research priorities in pediatric PN [51]. Ninety-nine research priorities were identified and were ranked in order of importance. Thirty-five of these priorities concerned energy, macronutrient and micronutrient supplementation in critically ill infants and children. Three topics received the highest research priorities. These were: 1. understanding the relationship between total energy intake, rapid catch-up growth and long-term metabolic function and neurocognitive outcomes, 2. defining optimal energy intake for different phases of illness (acute, stable and recovery) and the optimal route and doses of macro- and micronutrients and 3. defining the optimal dose and composition of amino acid mixture for optimal short- and long-term clinical outcomes. Identifying these knowledge gaps is important to further design studies, which could ultimately improve the outcome of critically ill children. Of utmost importance is to have a nutritional treating algorithm to decide whether to start with EN and PN and that can be used for daily clinical practice [52].

## 9. Conclusions

PN is generally indicated in infants and children who are unable to tolerate adequate oral or enteral feedings to sustain their nutritional requirements;In critically ill children, dosage of carbohydrates, amino acids and lipids is dependent on the phase of disease;The current international guidelines recommend considering withholding parenteral macronutrients during the first week of pediatric critical illness, while providing micronutrients;PN can induce adverse effects both in the short term and in the long term. The risk is reduced by a clear approach, the establishment of a multidisciplinary nutrition support team and the avoidance of unbalanced or excessive substrate supplies

## Figures and Tables

**Figure 1 nutrients-14-01819-f001:**
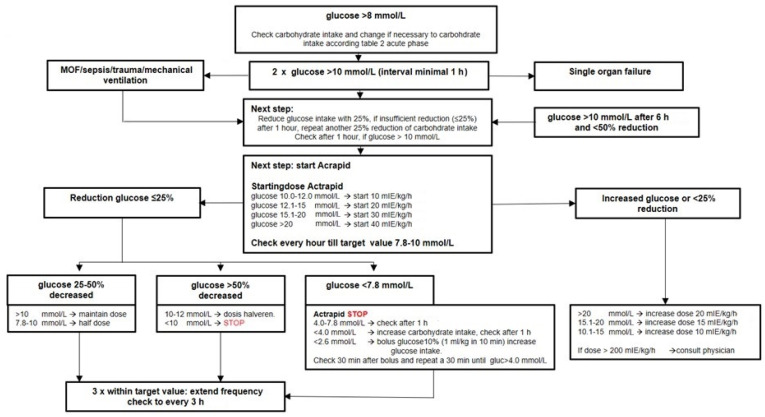
The insulin (Actaprid) protocol for treating hyperglycaemie (non-diabetes). Stop Actrapid if: doses < 5 mIE/kg/h or <20 mIE/kg/h. After 1 h control: if glucose ≤ 10 mmol/L, definitively stop Actaprid, otherwise start it again. Control glucose after stopping Actrapid: after 3, 6, 12 and 24 h. Points of attention: if at once enteral nutrition is started at 50% and glucose infusion is decreased with 50%, control glucose 3 times hourly; if continuous enteral nutrition is changed in intermittent administration, stop Actrapid; if any procedure is done, always continue glucose infusion and Actrapid; and if >50% oral nutrition, stop Actrapid. MOF, multiple organ failure.

**Figure 2 nutrients-14-01819-f002:**
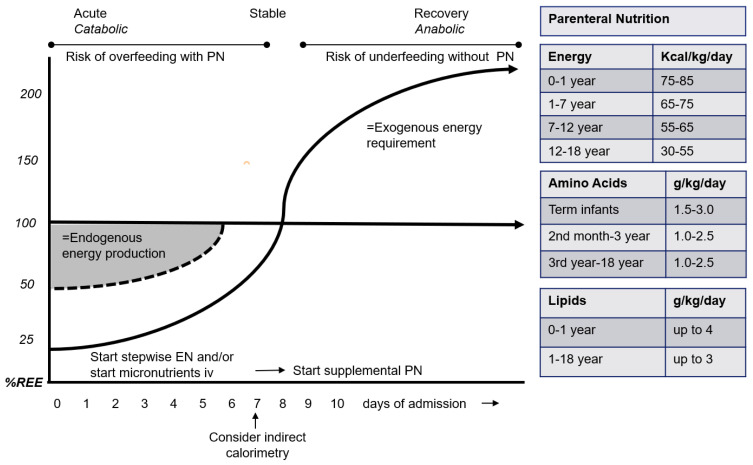
The dynamic energy needs during the different phases of critical illness, and the amount of parenteral nutrition (adapted from Ref. [36]). PN, parenteral nutrition; EN, enteral nutrition; REE, resting energy expenditure; iv, intravenous.

**Figure 3 nutrients-14-01819-f003:**
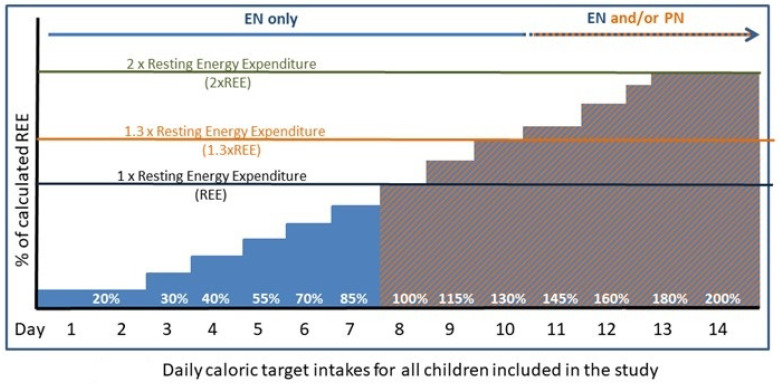
A stepwise approach to delivering calories with enteral and/or parenteral nutrition EN = enteral nutrition, PN = parenteral nutrition and REE = resting energy expenditure. The green line is to indicate intake (2 × REE) for neonates and infants. The purple line is to indicate the intake (1.3 × REE) for adolescents. The intake is based upon Holliday and Segar equations for maintenance fluid requirements [37].

**Table 1 nutrients-14-01819-t001:** The definitions of the three phases of the stress response in critically ill children (Ref. [7]).

	Definition
**Acute phase**	The first phase after the event, characterized by the requirement of (escalating) vital organ support. The phase when the patient requires vital organ support (sedation, mechanical ventilation, vasopressors and fluid resuscitation)
**Stable phase**	The stabilization or weaning of vital organ support, while the different aspects of the stress response are not (completely) resolved. The patient is stable on, or can be weaned off, this vital support
**Recovery phase**	Clinical mobilization with the normalization of neuro-endocrine, immunologic and metabolic alterations, characterized by a patient who is mobilizing

**Table 2 nutrients-14-01819-t002:** The carbohydrate (mg/kg/min) intake during the different phases of critical illness (Ref. [23]).

	Acute Phase	Stable Phase	Recovery Phase
Newborn	2.5–5	5–10	5–10
28 d–10 kg	2–4	4–6	6–10
11–30 kg	1.5–2.5	2–4	3–6
31–45 kg	1–1.5	1.5–3	3–4
>45 kg	0.5–1	1–2	2–3

**Table 3 nutrients-14-01819-t003:** The micronutrient provision in critically ill children (Ref. [29]).

Weight Class	Electrolyte Infusion	Vitamin and Trace Element Infusion
<5 kg *	Glucose 5%—NaCl 0.45% 113 mL, KCl 15% (2 mmol/mL) 0.8 mL, Ca-gluconate 10% (0.23 mmol/mL) 4 mL, Mg-sulphate 10% (0.4 mmol/mL) 0.50 mL and Glycophos (1 mmol P/mL, 2 mmol Na/mL) 1.2 mL 120 mL/kg/d, as continuous infusion	Soluvit^®^ (Fresenius Kabi, Rotterdam, The Netherlands) 1.5 mL/kg, Vitintra Infant^®^ (Fresenius Kabi, Rotterdam, The Netherlands) 2.5 mL/kg (max 10 mL), Peditrace^®^ (Fresenius Kabi, Rotterdam, The Netherlands) 1 mL/kg and NaCl 0.9% 37 mL
5–12 kg	Glucose 2.5%—NaCl 0.45% 69 mL, KCl 15% (2 mmol/mL) 0.6 mL, Ca-gluconate 10% (0.23 mmol/mL) 1.5 mL, Mg-sulphate 10% (0.4 mmol/mL) 0.25 mL and Glycophos (1 mmol P/mL, 2 mmol Na/mL) 0.5 mL 72 mL/kg/d, as continuous infusion	Soluvit^®^ (Fresenius Kabi, Rotterdam, The Netherlands) 1.5 mL/kg (max 8 mL), Vitintra Infant^®^ (Fresenius Kabi, Rotterdam, The Netherlands) 10 mL, Peditrace^®^ (Fresenius Kabi, Rotterdam, The Netherlands) 1 mL/kg (max 10 mL) and NaCl 0.9% 37 mL
12–30 kg	Glucose 2.5%—NaCl 0.45% 69 mL, KCl 15% (2 mmol/mL) 0.6 mL, Ca-gluconate 10% (0.23 mmol/mL) 1.5 mL, Mg-sulphate 10% (0.4 mmol/mL) 0.25 mL and Glycophos (1 mmol P/mL, 2 mmol Na/mL) 0.5 mL 72 mL/kg/d, as continuous infusion	Soluvit^®^ (Fresenius Kabi, Rotterdam, The Netherlands) 8 mL, Vitintra Infant^®^ (Fresenius Kabi, Rotterdam, The Netherlands) 10 mL, Supliven^®^ Fresenius Kabi, Rotterdam, The Netherlands0.25 mL/kg (max 10 mL) and NaCl 0.9% 50 mL
>30 kg	Glucose 2.5%—NaCl 0.45% 69 mL, KCl 15% (2 mmol/mL) 0.6 mL, Ca-gluconate 10% (0.23 mmol/mL) 1,5 mL, Mg-sulphate 10% (0.4 mmol/mL) 0.25 mL and Glycophos (1 mmol P/mL, 2 mmol Na/mL) 0.5 mL 48 mL/kg/d, as continuous infusion (max 2 L/d)	Soluvit^®^ (Fresenius Kabi, Rotterdam, The Netherlands) 8 mL, Vitintra Infant^®^ (Fresenius Kabi, Rotterdam, The Netherlands) 10 mL, Supliven^®^ (Fresenius Kabi, Rotterdam, The Netherlands) 0.25 mL/kg (max 10 mL) and NaCl 0.9% 50 mL

NaCl, sodium chloride; KCl, potassium chloride; Ca, calcium; Mg, magnesium; P, phosphorus; Na, sodium. * fluid intake at day 5 after birth.

**Table 4 nutrients-14-01819-t004:** The Schofield formulas for estimating resting energy expenditure (REE) from weight (kg) and height (m) in kcal/day * [42].

Age (year)	Boys	Girls
0–3 year	60.9 × weight − 54	61.0 × weight − 51
	0.167 × weight + 1516.7 × height − 617.6	16.2 × weight + 1022.7 × height − 413.5
3–10 year	22.7 × weight+ 495	22.5 × weight + 499
	19.6 × weight + 130.2 × height + 414.9	17.0 × weight + 161.7 × height + 371.2
10–18 year	17.5 × weight + 651	12.2 × weight (kg) + 746
	16.2 × weight + 137.1 × height + 515.5	8.4 × weight + 465.4 × height + 200.0

* 1 kcal = 4186 kJ.

## Data Availability

Not applicable.

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
