# Peer review of "PN Administration in Critically Ill Children in Different Phases of the Stress Response"

_nutrients, 2022, doi:10.3390/nu14091819_

Round 1

Reviewer 1 Report

This is an interesting study on an interesting topic. It is well writing article on PN Administration in Critically Ill Children in Different Phases of the Stress Response with minor revision.

 The minor comments

  • In the section of discussion, it is necessary to report all the new studies of the literature on this topic which is now missing. I recommend to be rewriting again in a good manner.
  • The conclusion of the study should be written purposeful, clear and conclusive.
  • Limitation of the study should be included.
  • The references; Revise again, there are some incomplete.

Author Response

  • We thank the reviewer for the comments and will answer to the raised questions and commentary point by point. All changed sentences and literature are yellow marked. 
  •  
  • 1. In the section of discussion, it is necessary to report all the new studies of the literature on this topic which is now missing. I recommend to be rewriting again in a good manner.
  • Answer: we have rewritten part of the discussion which is more a consideration than a discussion because in a review normally there is no discussion about the review itself
  • 2. The conclusion of the study should be written purposeful, clear and conclusive.
  • Answer: we have added conclusions at the end of the discussion
  •  
  • 3. Limitation of the study should be included.
  • Answer: because this is a review and not an original study we can't write limitations
  • 4.The references; Revise again, there are some incomplete.
  • Answer: we have checked all the references, revised some of them and added some new references

Attached the revised manuscript

Reviewer 2 Report

The paper was well written and interesting. It clearly summarized new evidence in the field of parenterale nutrition for critically ill children.

I have no further suugestions for the paper. 

Author Response

We would like to thank the reviewer for the response